# Cost Deviation Model of Construction Projects in Saudi Arabia Using PLS-SEM

Abdullah M. Alsugair

Department of Civil Engineering, College of Engineering, King Saud University,
P.O. Box 800, Riyadh 11421, Saudi Arabia; amsugair@ksu.edu.sa

**Abstract:** Miscalculations during cost estimation can have adverse effects on construction projects, including delaying or canceling planned projects, reducing project scope, and creating considerable financial risks for both owners and contractors. The objective of this research was to identify the major factors that cause cost deviation and study the effect of the interaction between these factors on cost deviation prior to the tender phase of construction projects in the Kingdom of Saudi Arabia. This was accomplished by carrying out a comprehensive literature review of the factors affecting cost deviation; implementing a survey questionnaire for project participants, including contractors, consultants, and clients, who are aware of the construction industry in Saudi Arabia; and developing a model for cost deviation based on the questionnaire data using the partial least square structural equation model (PLS-SEM). The cost deviation model was developed, and the PLS-SEM provided the critical factors affecting cost deviation and gave theoretical support for the study's conceptual framework. The results revealed that the problem is shared by the owners and contractors, as the factors with the highest rankings were project characteristics, contractual procedures, and estimator performance. The study also showed that the issue of cost deviation is more important to owners than contractors, as the predictive relevance of project characteristics, contractual procedures, and estimator performance were 0.229, 0.335, and 0.197, respectively, for the client–consultant model, and 0.117, 0.118, and 0.292, respectively, for the contractor model. The results indicate the need to control the highest-ranked factors to enhance the efficiency of the cost estimation process. This study contributes to the body of knowledge by generating the PLS-SEM model that takes into account the indirect relationships among affecting cost deviation factors and considers these relationships while preparing the bid to reduce the deviation cost.

**Keywords:** model; factors; cost; project; contract; hypothesis; reliability; deviation; clients

## 1. Introduction

Cost estimation is an essential task for the project initiation phase and has a substantial effect on project performance. The problematic issue that arises from cost estimation is the variation between the contract amount and the estimated project cost. This difference is significant when calculating cost deviation. Chung and Ashuri [1] defined cost deviation as the ratio of the variation between the estimated project costs and the winning bid to the latter value.

Significant cost deviation can generate financial risks for clients and contractors. For example, upward deviation normally causes project cancellation and the inappropriate allocation of public funds. Downward deviation, on the other hand, results in significant cost overruns for both owners and contractors because of claims and modifications that occur during construction. The cost deviation of 25% of the submitted bids for highway projects from the Washington State Department of Transportation (WSDOT) between 2006 and 2011 was greater than 10% [2]. According to a study carried out by Flyvbjerg et al. [3], cost variation has not decreased during the past 70 years. Based on information from the Office of Government Accountability (1997), the percentage of cost increase for highway

projects is 77% in the United States. In another study, it was observed that the cost overrun is 20% for road projects and 90% for transportation infrastructure projects [4].

Cost variation can have detrimental impacts, including the delay or termination of projects, scope reduction, and a loss of public trust [5]. Therefore, to avoid the misappropriation of government funds and produce the appropriate economic outcome for the general public, a proper explanation of cost variation is crucial [6]. An effective method to capture cost deviation is to investigate the causes that influence that issue. According to a study by Ibrahim et al. [7], cost estimates are enhanced considerably by clear and detailed project documentation, specifications, and drawings; the expertise of the cost estimator; the detail, quality, and accuracy of cost information; the availability, price, and quality of materials; and prior experience with comparable projects. Saqer et al. [8] identified the major factors affecting the preciseness of pretender estimates for 67 worldwide projects. They addressed 45 factors and classified them into 11 categories. The major factors included process design, team knowledge, cost data, time available to prepare estimates, site needs, and bidding and labor climate. The risk factors for cost variation connected to contractors were highlighted by Mahamid [9] as changes in currency exchange rates, project funding, contract management, the level of competition, and material cost. Another study utilized a feedforward neural network model to enhance clients' estimates [10]. This study was based on the Friedman principle, which states that maximizing the predicted markup size and recasting it as an optimization problem yields the lowest construction bid [11]. In terms of the problem in Saudi Arabia, Mahamid et al. [12] carried out a questionnaire survey and statistically analyzed 44 factors affecting cost deviation to establish the major causes of cost variance in building projects in Saudi Arabia based on the client's perspective. The authors reported that the significant factors were the level of competitors submitting their bids, the fluctuation of material price, interactions between suppliers and the client, and the estimation approach used. On the other hand, the timing of advertisement, site conditions, equipment prices, material scarcity, and the category of project had a minimum influence on cost deviation.

Based on our review of the literature, there have been few studies conducted in Saudi Arabia and none has considered the interaction influence among the factors affecting cost deviation. Thus, the objective of this study was to develop three models to determine the main factors affecting cost deviation as well as the interactions that have an effect on those elements from three perspectives: client–consultant, contractor, and common (client–consultant and contractor). This objective was achieved by the following steps: (1) review the literature to list the cost deviation factors; (2) design and carry out a questionnaire survey; (3) use PLS-SEM to evaluate the questionnaire data, utilizing the SmartPLS program; and (4) interpret the output. The main components of this paper are shown in Figure 1.

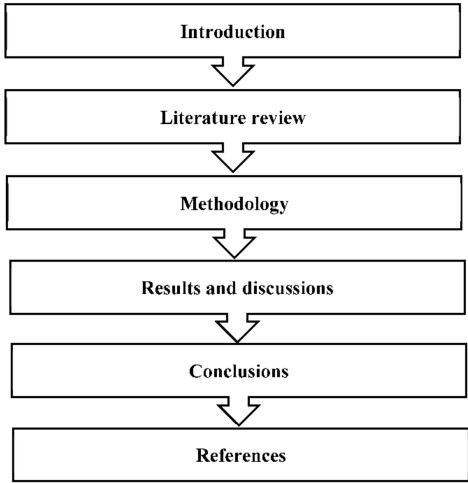

**Figure 1.** Research flow chart.

## 2. Literature Review

The construction sector has one of the highest annual rates of company failures and related liabilities when compared to other sectors [13,14]. Several uncertainties and risks should be attended to and expected by construction management. Cost deviation, which is the ratio of the difference between a contract amount and bid to the contract amount, is one of these risks. Numerous studies examining the factors driving cost deviation have been conducted to better recognize and understand the issue of cost variance.

Flyvbjerg et al. [4] conducted a study to identify the influence of project features on cost variance. In this study, they investigated the cost variation for 258 highway projects in various countries. They stated that cost deviation is a universal phenomenon and has not decreased over the last seven decades. Memon et al. [15] carried out a statistical analysis of the different factors influencing Malaysian project costs among project consultants. They demonstrated that bad site management and supervision by contractors were strongly positively correlated with poor planning and scheduling by contractors, the late delivery of materials and equipment, and the connection between management and labor. Inaccurate planning and scheduling were strongly positively correlated with contractors' lack of experience. Ibrahim and Elshwadfy [7] identified and ranked the factors affecting the preciseness of estimating the cost of construction projects in Egypt using a questionnaire survey. The respondents were 106 consultants and contractors.

Baek et al. [16] identified factors affecting cost deviation using regression analysis. An analytical model was constructed using historical cost data for highway projects awarded in the state of Louisiana between 2011 and 2015. The authors concluded that the quantity of contract activities, the value of paving projects, the intensity of bidding competition, and the price of crude oil all had a significant impact on cost variance. Faten Albtoush et al. [17] reviewed the aspects affecting the estimated cost of construction projects in different countries. They concluded that the significant factors influencing cost estimation in construction projects in most countries were a short preparation duration; the accuracy and reliability of the cost information; the clarity of the details in the drawings and specifications; the experience of the estimators; the type of construction; and the location of the project. Li et al. [10] examined the variables influencing the client's estimate accuracy for highway projects and developed a method to forecast the low bid using a time series and the owner's estimate.

In terms of research carried out in Saudi Arabia, Bubshait and Al-Juwairah [18] examined the importance of 42 factors affecting construction costs in Saudi Arabia by measuring the level of importance and ranking them based on their severity index. They concluded that incorrect planning, contractor experience, contract management, and financial difficulties were the most critical factors affecting construction cost. Shash and Ibrahim [19] statistically examined consultant firms that compute the pretender cost estimates of housing buildings in Saudi Arabia. Their results revealed that few firms use specialized software packages in conducting estimating services, implement a structured approach when accounting for design variables, or develop models created by construction researchers.

Previous research has illustrated that cost deviation has a significant impact on construction projects, and most studies have focused on identifying the main affecting factors. However, they do not consider the interrelationships among the factors (indirect influences) and do not examine the impact of these indirect influences on cost deviation. A structural equation model (advanced statistical analysis) may examine the complex interrelationships among groups.

There are two common structural equation models utilized in research: CB-SEM and PLS-SEM. Selecting the appropriate model depends upon the purpose of the research and sample size; the CB-SEM model is used for confirmatory research and it requires a large sample, while the PLS-SEM is suitable for developing exploratory studies. PLS-SEM can investigate the complex interrelationships between group factors. In addition, Willaby et al. [20] and Kock [21] stated PLS-SEM's capability to accommodate analysis in small sample size experiments with unbiased results. Hence, the merits of PLS-SEM were

utilized in the construction industry to examine the influence of different potential factors on the cost deviation and test the relationship among these factors.

In this study, data collected from a questionnaire survey carried out in Saudi Arabia were analyzed and PLS-SEM was used to develop three models (client–consultant, contractor, and common). The proposed models aim to provide the main affecting factors and recognize the influence of the interaction among these factors to establish indirect influences on cost deviation.

### 3. Methodology

The methodology of this study consisted of reviewing and collecting data on the factors affecting cost deviation, designing a questionnaire, and analyzing the questionnaire data to create a PLS-SEM model using the SmartPLS program. The main features of the methodology are shown in Figure 2.

**Figure 2.** Methodology flow chart.

*3.1. Collecting the Factors Affecting Cost Deviation*

The procedures adopted for selecting factors were chosen based on a review of the literature. Through a thematic review, it was observed that many factors had synonymous notation in different reports. Therefore, similar factors were abbreviated, refined, and linked with their related category. The selected cost deviation factors were classified into the following seven groups: scope quality, information quality, estimator performance, external factors, contractor organization, contractual procedures, and project characteristics. The groups with their factors and symbols are shown in Table 1 [7,8,12,14,15,17,22,23]. These seven groups represent the main components of the PLS-SEM.

**Table 1.** Main groups and their effect factors.

| Group | Factor | Factor Description |
|---|---|---|
| Scope quality (SQ) | SQ1 | Definition of the scope is clear to the client. |
| | SQ2 | Previous client/consultant experience with contract. |
| | SQ3 | Level of team integration. |
| | SQ4 | Buildability of design. |
| | SQ5 | Having a number of other projects with the client. |
| | SQ6 | Client's level of involvement in the estimation procedure. |
| | SQ7 | Client's priority level of the project. |
| Contractor organization (CO) | CO1 | Management team (suitability, experience, and performance). |
| | CO2 | Financial capability. |
| | CO3 | Past relationship with a client. |
| | CO4 | Past loss/profit in similar projects. |
| | CO6 | Payment record of a client. |
| | CO7 | Relationship with subcontractors and suppliers. |
| | CO8 | Current workload. |
| | CO9 | Experience with similar projects. |
| Estimator performance (EP) | EP1 | Experience in pricing construction. |
| | EP2 | Impact of team integration and alignment. |
| | EP3 | Estimation method. |
| | EP4 | Time allowed for preparing cost estimates. |
| | EP5 | Number of estimating staff. |
| | EP6 | Estimator's workload during estimation. |
| | EP7 | Application of alternative methods by an organization. |
| Information quality (IQ) | IQ1 | Completeness of cost information. |
| | IQ2 | Reliability and accuracy of cost information. |
| | IQ3 | Detailed and clear specifications and drawings. |
| Project characteristics (PC) | PC1 | Type/function of the building. |
| | PC2 | Size/gross floor area. |
| | PC3 | Type of structures (steel, concrete, or masonry). |
| | PC4 | Quality of finishing. |
| | PC5 | Priority of project. |
| | PC6 | Duration of project period. |
| | PC7 | Inadequate production of raw materials for the country. |
| | PC8 | Client's financial situation and budget. |
| | PC9 | Site condition in terms of site accessibility, site topography, site requirements, and the level of uncertainty in soil conditions. |

**Table 1.** *Cont.*

| Group | Factor | Factor Description |
|---|---|---|
| External factors (EF) | EF1 | Material cost. |
| | EF2 | Labor cost. |
| | EF3 | Equipment cost. |
| | EF4 | Overhead costs. |
| | EF5 | Government requirements (permits). |
| | EF6 | Weather. |
| | EF7 | Lack of information and coordination between government agencies. |
| | EF8 | Price fluctuation. |
| | EF9 | Inflationary pressure. |
| | EF10 | Economic insatiability. |
| | EF11 | Currency exchange. |
| | EF12 | Taxation on imported materials. |
| | EF13 | Monopoly. |
| Contractual procedures (CP) | CP1 | Client's evaluation and awarding policy. |
| | CP2 | Contract conditions. |
| | CP3 | Number of bidders on competitive projects. |
| | CP4 | Allowed contingencies. |
| | CP5 | Tendering duration. |
| | CP6 | Duration between the announcement of the project, submission of the offer, and awarding of the contract. |
| | CP7 | Availability of other projects for tendering. |
| | CP8 | Performance bond and warranty arrangements. |
| | CP9 | Amount of specialist work. |

*3.2. Questionnaire Design and Implementation*

A questionnaire was utilized to gather the relevant information in order to research and pinpoint Saudi Arabian cost deviation factors. The participants represented project contributors and included clients, consultancy offices, and contractors. The survey structure was designed to provide two main types of information: general information (demographic information and main cost estimation practice) and factors affecting cost deviation.

The influence of factors in the scope quality group on cost deviation was apparent mostly during the design stage. Therefore, the ranking of scope quality factors was limited to client and consultant participants. Similarly, the influence of factors belonging to the contractor organization category on construction cost estimation was apparent mostly during the tendering stage. Therefore, the ranking of contractor organization factors was limited to contractor participants. The response system used to record the participants' perceptions about the factors affecting the accuracy of estimation was based on a five-point Likert scale, where one denoted "totally disagree" and five denoted "totally agree".

*3.3. Statistical Analysis (Three Models) Using PLS-SEM (SmartPLS Program)*

Based on the questionnaire results, the factors affecting cost deviation were studied directly by testing the impact of each group on cost deviation and indirectly by testing the relationships among the groups using the three different viewpoints (client–consultant, contractor, and common). The proposed structure of the relationships among factors related to cost estimation accuracy was modeled using structural equation modeling with partial least square procedures. Three models were constructed: a client and consultant model ($Model_{client-consultant}$), contractor model ($Model_{contractor}$), and common model ($Model_{common}$), as shown in Figure 3. Each model consists of an inner model and an outer model. The outer model represents the relationships between groups (constructs) and their

factors affecting cost deviation (indicators) and is called a measurement model. The inner model, however, represents the relationships among the groups (constructs) and it is called a structural model. The groups and their factors affecting cost deviation are illustrated in Table 1. Figure 4 shows the common model: the outer model is represented by the light blue region, while the inner model is shown in the light red region. The groups are represented by blue circles, while the factors (indicators) are represented by yellow rectangles. It should be noted that some factors were omitted due to assessment processing, which will be illustrated in the following sections. The SmartPLS software was used to obtain the model's results. The proposed structure was modeled and showed the relationships among groups related to cost estimation accuracy from client–consultant, contractor, and combined points of view. The hypotheses associated with each of the three models are illustrated in Table 2.

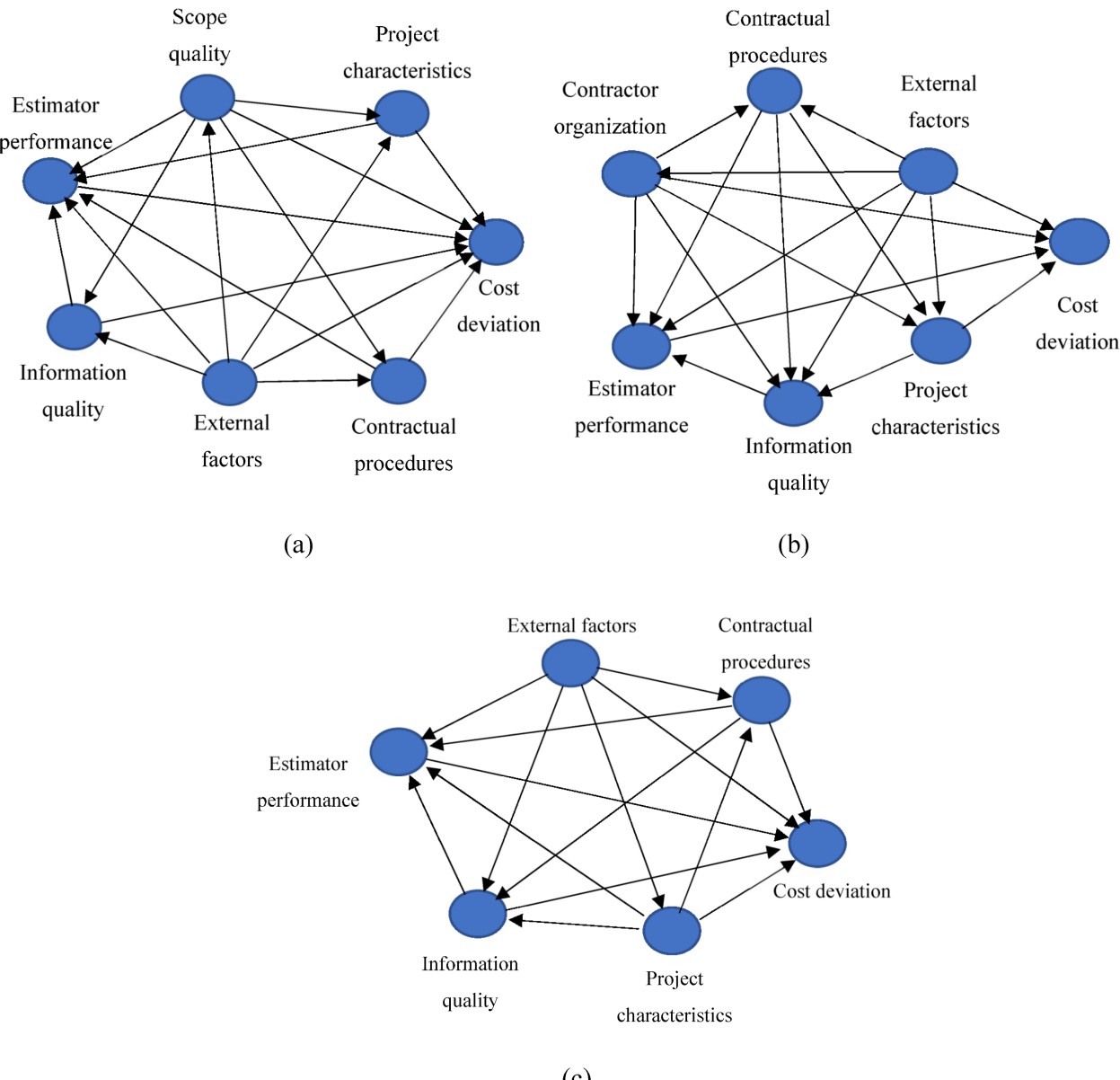

**Figure 3.** The three models: (**a**) client–consultant model, (**b**) contractor model, (**c**) common model.

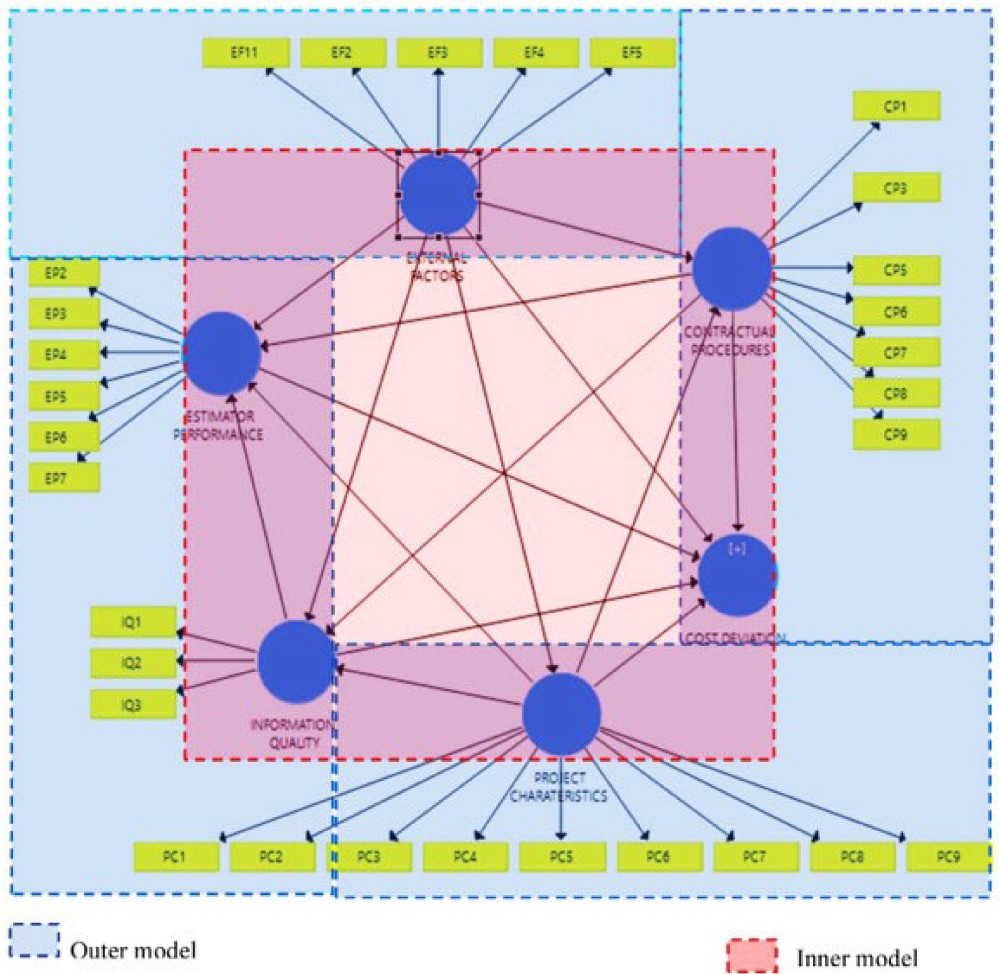

**Figure 4.** Outer and inner model of the common model.

**Table 2.** Hypotheses for the three models.

| Model | Symbol | Hypothesis |
|---|---|---|
| $Model_{client-consultant}$ | H1 | Contractual Procedure => Cost Deviation |
| | H2 | Contractual Procedure => Estimator Performance |
| | H3 | Estimator Performance => Cost Deviation |
| | H4 | External Factors => Contractual Procedure |
| | H5 | External Factors => Cost Deviation |
| | H6 | External Factors => Estimator Performance |
| | H7 | External Factors => Information Quality |
| | H8 | External Factors => Project Characteristics |
| | H9 | External Factors => Scope Quality |
| | H10 | Information Quality => Cost Deviation |
| | H11 | Information Quality => Estimator Performance |
| | H12 | Project Characteristics => Cost Deviation |
| | H13 | Project Characteristics => Estimator Performance |
| | H14 | Scope Quality => Contractual Procedure |
| | H15 | Scope Quality => Cost Deviation |
| | H16 | Scope Quality => Estimator Performance |
| | H17 | Scope Quality => Information Quality |
| | H18 | Scope Quality => Project Characteristics |

**Table 2.** *Cont.*

| Model | Symbol | Hypothesis |
|---|---|---|
| *Model_contractor* | H1 | Contractor Organization => Contractual Procedures |
| | H2 | Contractor Organization => Cost Deviations |
| | H3 | Contractor Organization => Estimator Performance |
| | H4 | Contractor Organization => Information Quality |
| | H5 | Contractor Organization => Project Characteristics |
| | H6 | Contractual Procedures => Estimator Performance |
| | H7 | Contractual Procedures => Information Quality |
| | H8 | Contractual Procedures => Project Characteristics |
| | H9 | Estimator Performance => Cost Deviations |
| | H10 | External Factors => Contractor Organization |
| | H11 | External Factors => Contractual Procedures |
| | H12 | External Factors => Cost Deviations |
| | H13 | External Factors => Estimator Performance |
| | H14 | External Factors => Information Quality |
| | H15 | External Factors => Project Characteristics |
| | H16 | Information Quality => Estimator Performance |
| | H17 | Project Characteristics => Cost Deviations |
| | H18 | Project Characteristics => Information Quality |
| *Model_common* | H1 | Contractual Procedures => Cost Deviation |
| | H2 | Contractual Procedures => Estimator Performance |
| | H3 | Contractual Procedures => Information Quality |
| | H4 | Estimator Performance => Cost Deviation |
| | H5 | External Factors => Contractual Procedures |
| | H6 | External Factors => Cost Deviation |
| | H7 | External Factors => Estimator Performance |
| | H8 | External Factors => Information Quality |
| | H9 | External Factors => Project Characteristics |
| | H10 | Information Quality => Cost Deviation |
| | H11 | Information Quality => Estimator Performance |
| | H12 | Project Characteristics => Contractual Procedures |
| | H13 | Project Characteristics => Cost Deviation |

The three models were evaluated in terms of measurement and structural assessments. The assessment of measurement consisted of convergent reliability and discriminant validity. The structural assessment examined the relationships (hypothesis testing) and evaluated the strength of the relationships using the determination coefficient ($R^2$), path coefficients (w), prediction relevance ($Q^2$), and model fit.

### 3.4. Assessment of the Outer Model (Measurement Model)

This assessment comprised convergent validity and discriminant validity. The purpose of convergent validity is to examine the factors that have a favorable impact on a group, for example, to check for correlations between the influencing factors SQ1–SQ7 and the scope quality group. Convergent validity was carried out using construct reliability, which includes the composite reliability, Cronbach's alpha, and average variance extracted for each group with their affecting factors. Discriminant validity is the degree to which a group—such as SQ, PC, CP, EP, EF, or IQ—differs from one another. Therefore, proving discriminant validity suggests that a concept is distinct and includes phenomena not covered by other constructs in the model [24]. The Fornell–Larcker criterion, which was used for the groups, and cross loading, used for the influencing factors, were the two

criteria that were utilized to obtain the discriminant validity requirements. Each model consisted of different groups that had several of the factors affecting cost deviation.

In terms of convergent validity, the construct reliability coefficients are functions of the outer loading of the affecting factors. To examine the significance of factors in their groups, the outer loading of a factor (*l*) was studied. If *l* was more than 0.7, the factor was retained in the model. On the other hand, when *l* ranged from 0.4 to 0.7, the influence of the deletion of the factor on the composite reliability of the group factor was studied. A factor was deleted if the deletion led to an increase in the composite reliability of the group. However, when there was no increase in the composite reliability of the group due to the deletion of the factor, the factor was retained in the model. These steps are shown in Figure 5. The Cronbach's $\alpha$ and composite reliability ($\rho_c$) were computed using Equations (1) and (2) [25], respectively, as follows:

$$Cronbach's\ \alpha = \left(\frac{M}{M-1}\right)\left(1 - \frac{\sum_{i=1}^{M} s_i^2}{s_t^2}\right) \tag{1}$$

$$\rho_c = \frac{\left(\sum_{i=1}^{M} l_i\right)^2}{\left(\sum_{i=1}^{M} l_i\right)^2 + \sum_{i=1}^{M} var(e_i)} \tag{2}$$

where $s_i^2$ is the variance of factor *i*; $s_t^2$ refers to the variance associated with the observed total factors of a specified group; $l_i$ is the standardized outer loading of factor *i* of a specific group; *M* is the total number of factors in a specific group; and $var(e_i)$ is the variance of measurement errors.

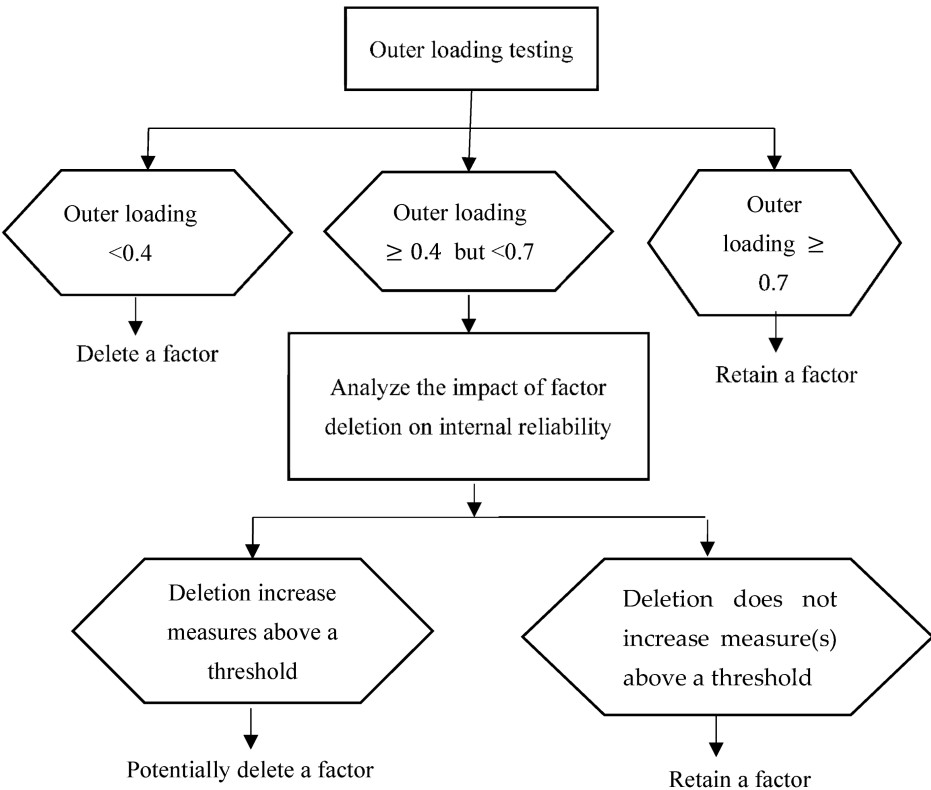

**Figure 5.** Outer loading testing [24].

In addition, the reliability of any group in a model in terms of the average variance extended (*AVE*), which was computed using Equation (3), was evaluated and shall be made higher than the threshold value (>0.5) by omitting the factors with a lower outer loading (*l*) [25].

$$AVE = \frac{\sum_{i=1}^{M} l^2}{M} \tag{3}$$

According to Hair et al. [24], an *AVE* of less than 0.5 implies that more variance is still accounted for by the factors' errors than by the other group.

Discriminant validity examines the distinct validity of one group to another. It is determined by evaluating cross-loading and the Fornell–Larcker criterion. Cross-loading was used to assess and evaluate the discriminant validity of the factors. The value of the cross-loading of the factors on their group should be more than the cross-loading on other groups. To check the discriminant validity of the model's groups, the Fornell–Larcker criterion was used, which is the square root of the group's AVE and should be greater than its highest correlation with any other group. The principle of the Fornell–Larcker method is based on the idea that a group's factor shares more variance with its associated factors than with another group [24].

*3.5. Assessment of the Inner Model*

After assessing the outer model of the three models, the inner models (structure models) were examined for collinearity. Since the estimation of path coefficients in the structural models was based on Ordinary Least Square (OLS), regressions were performed for each factor variable on its corresponding predecessor constructs. To check for collinearity, the variance influence factor (VIF) of each factor was determined and the value had to be lower than five. Otherwise, the factor was considered for elimination. The structural assessments were also considered, including path coefficients and hypothesis tests for the relationships among groups and the determination coefficients of the prediction relevance of the groups.

The hypothesis tests were carried out by determining one of two coefficients, the t-value, or *p*-value. These coefficients depend on the standard error obtained by bootstrapping, which is generated by the SmartPLS program. The conclusion of the test hypothesis depends on either the t-value or *p*-value. In terms of the t-value, when the t-value surpasses the critical value, the coefficient is regarded as statistically significant at a specific error probability (i.e., significance level). Researchers typically use a significance level of 5% when studying marketing. Therefore, the significance level was set at 5% with a critical t-value of 1.96 in this study. When using the *p*-value, the relationship under examination is significant at a 5% level, and the *p*-value needs to be lower than 0.05 when assuming a significance threshold of 5%.

The relation between the path coefficients of the structural model is explicable. A path coefficient's impact on the endogenous group variable is stronger if it is larger than another path coefficient. More specifically, the standardized beta coefficients in an ordinary least square (OLS) regression can be used to represent each path coefficient in the route model: the exogenous group changes by one unit. If the path coefficient is statistically significant, its value reveals how closely the two groups are related.

The determination coefficient ($R^2$) is an important assessment in the evaluation of the structural model. It denotes the quantity of variance in the dependent group explained by all of the groups that were linked to the dependent group [26,27]. It is challenging to decide on rules of thumb for acceptable $R^2$ values. Hair et al. [24] pointed out that the $R^2$ values of 0.25, 0.5, or 0.75 for the dependent group represent weak, moderate, or strong predictive power, respectively. On the other hand, many researchers use prediction relevance ($Q^2$) to measure the predictive power of a model.

## 4. Results and Discussion

### 4.1. Questionnaire Results

The targeted sample comprised 173 participants. The sample was divided as follows: 112 participants from the client/consultant group and 61 participants from the contractor group. The targeted sample actively participated in answering the research questionnaire within the given time limit.

The questionnaire results suggested a common agreement between contractor and client-consultancy participants on the possible effect of some factors on cost deviation. The rankings of some group factors, such as contractual procedures, estimator performance, project characteristics, and materials, were high compared to the remaining categories, including external factors and quality of information. According to the perceptions of the participants, the questionnaire results suggested the existence of factors that could critically affect cost deviation, which needed to be further analyzed and investigated.

The presence of a cost estimating unit within companies and stakeholder satisfaction with construction cost estimation represent proof of the general acceptance of this result. The awarded contract value was satisfactory for about 70% of the survey participants in the main contractors group who had already declared that they possessed a cost estimation unit. A cost estimation unit was not present in 11.92% of organizations, although 40% of clients and consultants were satisfied with building cost estimation. The effects of a lack of estimation units during the design development stage will place a major risk on budget allocation for projects, which will ultimately have an impact on how much a project will cost throughout its life cycle.

### 4.2. PLS-SEM Models Results

The results of the statistical analysis of the three models in terms of their outer (measurement) and inner (structure) models are discussed below.

#### 4.2.1. Outer Model Assessment

The following sections present the results of the convergent validity and discriminant validity assessments.

Convergent Validity

Table 3 shows the significant factors that satisfied the reliability and discriminately validity criteria and were retained in the three models. The three models shared the same view on the factors from the project characteristics (PC) and information quality (IQ) groups. Shash and Ibrahim [19] stated the importance of project size (PC2) on the cost deviation. They demonstrated how unexpected factors could affect the cost accuracy of a large project. The results of "information quality" factors are supported by [12,17,19,27,28], who state that the wrong estimation has a detrimental effect on cost accuracy. Furthermore, the three models were in agreement in terms of the contractual procedure (CP) factors. They were only different with regard to CP3, where CP3 was not significant in the contractor model. The importance of CP1, which represents the client's evaluation and awarding policy, having an influence on CD for the three models is supported by Ibrahim [14], Azhar et al. [28], and Mohamid and Aichouni [12], who stated that awarding policy has an influence on accuracy in cost estimating. In addition, Mohamid and Aichouni [12] pointed out the influence of CP3 on CD. In addition, the study carried out by [19] confirmed the influence of tendering duration (CP5) on the cost deviation; their results coincide with the findings of this paper. On the other hand, the factors from the estimator performance (EP) and external factors (EF) groups were not identical in the three models. For EP factors, the client–consultant model was more conservative in considering the factors to be as significant as those in the contractor model, while the common model was intermediate between the contractor and client–consultant models. The client–consultant model considered EP2, EP4–EP7, and EP1–EP6 as significant factors for the contractor model. The common model was the same as the client–consultant model with respect to the EP3 factor. The common

model considers the influence of EP2 through EP7. Azhar et al. [28] and Mohamid and Aichouni [12] confirmed that the influence of estimator experience (EP3) and estimating method (EP7) on the accuracy of the quantity computation of the bid was significant. These factors are considered in the common model. In terms of external factors, the three models considered EF2, EF4, and EF5 as significant factors. In addition to the previous factors, the contractor model also considered EF1, EF3, EF8, EF9, and EF10 to be significant factors for the contractor model and EF11 for both client–consultant model and the common model. The importance of labor costs (EF3) for cost deviation and accuracy costs was supported [12]. They stated that labor costs are highly affected by market conditions and consequently influence the accuracy of the bid cost and cost deviation. The main factors in the scope quality group for the client–consultant model were SQ5 and SQ7. Moreover, all contractor organization factors showed significant effects and were retained in the contractor model, except for CO1, CO4, and CO5.

**Table 3.** Factors affecting cost deviation that satisfied the measurement assessment for the three models.

|  | IQ | EP | EF | CP | PC | CO | SQ |
|---|---|---|---|---|---|---|---|
| *Model* <sub>client-contractor</sub> | IQ1 | EP2 | EF2 | CP1 | PC1 |  | SQ5 |
|  | IQ2 | EP4 | EF4 | CP3 | PC2 |  | SQ7 |
|  | IQ3 | EP5 | EF5 | CP5 | PC3 |  |  |
|  |  | EP6 | EF11 | CP6 | PC4 |  |  |
|  |  | EP7 |  | CP7 | PC5 |  |  |
|  |  |  |  | CP8 | PC6 |  |  |
|  |  |  |  | CP9 | PC7 |  |  |
|  |  |  |  |  | PC8 |  |  |
|  |  |  |  |  | PC9 |  |  |
| *Model* <sub>contractor</sub> | IQ1 | EP1 | EF1 | CP1 | PC1 | CO2 |  |
|  | IQ2 | EP2 | EF2 | CP5 | PC2 | CO3 |  |
|  | IQ3 | EP3 | EF3 | CP6 | PC3 | CO6 |  |
|  |  | EP4 | EF4 | CP7 | PC4 | CO7 |  |
|  |  | EP5 | EF5 | CP8 | PC5 | CO8 |  |
|  |  | EP6 | EF8 | CP9 | PC6 | CO9 |  |
|  |  |  | EF9 |  | PC8 |  |  |
|  |  |  | EF10 |  | PC9 |  |  |
| *Model* <sub>common</sub> | IQ1 | EP2 | EF2 | CP1 | PC1 |  |  |
|  | IQ2 | EP3 | EF3 | CP3 | PC2 |  |  |
|  | IQ3 | EP4 | EF4 | CP5 | PC3 |  |  |
|  |  | EP5 | EF5 | CP6 | PC4 |  |  |
|  |  | EP6 | EF11 | CP7 | PC5 |  |  |
|  |  | EP7 |  | CP8 | PC6 |  |  |
|  |  |  |  | CP9 | PC7 |  |  |
|  |  |  |  |  | PC8 |  |  |
|  |  |  |  |  | PC9 |  |  |

Table 4 shows the measurement model results along with the assessment criteria thresholds for the reliability assessment, including the average variance extracted (AVE), composite reliability, and Cronbach's alpha, for the three models. The threshold values of average variance extracted, composite reliability, and Cronbach's alpha were 0.5, 0.7, and 0.7, respectively.

**Table 4.** Summarized values for the convergent validity of the three models.

| | $Model_{Client-consultant}$ | | | $Model_{Contractor}$ | | | $Model_{Common}$ | | |
|---|---|---|---|---|---|---|---|---|---|
| | Cronbach's $\alpha$ | Composite Reliability | AVE | Cronbach's $\alpha$ | Composite Reliability | AVE | Cronbach's $\alpha$ | Composite Reliability | AVE |
| CP | 0.898 | 0.920 | 0.623 | 0.812 | 0.864 | 0.516 | 0.891 | 0.915 | 0.605 |
| PC | 0.781 | 0.851 | 0.541 | 0.870 | 0.899 | 0.533 | 0.890 | 0.912 | 0.539 |
| EP | 0.781 | 0.851 | 0.541 | 0.861 | 0.896 | 0.591 | 0.825 | 0.873 | 0.535 |
| EF | 0.727 | 0.829 | 0.555 | 0.872 | 0.897 | 0.531 | 0.782 | 0.851 | 0.537 |
| IQ | 0.701 | 0.830 | 0.620 | 0.834 | 0.902 | 0.756 | 0.764 | 0.864 | 0.680 |
| SQ | 0.601 | 0.833 | 0.714 | | | | | | |
| CO | | | | 0.812 | 0.864 | 0.516 | | | |

Generally, both the composite reliability and Cronbach's alpha of the latent variables were between 0.701 and 0.915, which was higher than the minimum level accepted (0.7) for the three models. An exception was found for scope quality (SQ) in the client–consultant model, with a Cronbach's alpha of 0.601. This low Cronbach's alpha value can be justified by the low number of indicators representing the latent variable (which contained two factors only). Since Cronbach's alpha is vulnerable to the number of factors (number of indicators) and underestimates the reliability of internal consistency, composite reliability is recommended to assess the consistency reliability of groups [22]. Consequently, the consistency reliability of the groups for each model was satisfied. The extended average variance values ranged from 0.516 to 0.714, which was more than the threshold value (0.5).

Discriminant Validity

Concerning the discriminant validity, Table 5 shows the Fornell–Larcker criterion results for the three models. The diagonal value, which relates to the average variance extracted (AVE), was larger than the remaining values, which represent the correlation between the groups. Comparing the values obtained, the results suggested that the latent variables were different from each other, as each group explained the variance of its factors instead of the variance of other categories. A given group should explain the variance of its indicators (factors) better than other groups. Consequently, the correlations with other latent constructs should be less than the square root of each construct's AVE. The cross-loading determined the correlation coefficient between each factor and its related group. It can be observed that the cross-loading associated with the factors in their related group was greater than all the cross-loadings that appeared in the remaining groups. Therefore, the cross-loading criteria related to discriminant validity were fulfilled for the three models.

The study carried out by [17] summarized the affecting cost deviation factors for projects that were constructed in New Zealand, Nigeria, Peninsular Malaysia, and Gaza Strip. It revealed that the most affecting factors were tendering duration (CP5), site condition (PC9), details, clear specification and drawings (IQ3), and experience estimator (EP1). Table 1 includes these factors, in which Table 5 represents the final outcome of the outer models' assessments of the three models. Therefore, the results of this study were confirmed by the different case studies reported in the study conducted by [17].

**Table 5.** Fornell–Larcker criterion results.

| | $Model_{Client\ consultant}$ | | | | | |
|---|---|---|---|---|---|---|
| | CP | EP | EF | IQ | PC | SQ |
| CP | 0.789 | | | | | |
| EP | 0.513 | 0.736 | | | | |
| EF | 0.706 | 0.512 | 0.745 | | | |
| IQ | 0.250 | 0.432 | 0.296 | 0.787 | | |
| PC | 0.529 | 0.389 | 0.659 | 0.203 | 0.739 | |
| SQ | 0.603 | 0.436 | 0.552 | 0.162 | 0.474 | 0.845 |

**Table 5.** *Cont.*

| | CO | CP | EP | EF | IQ | PC |
|---|---|---|---|---|---|---|
| | | | $Model_{Contractor}$ | | | |
| CO | 0.718 | | | | | |
| CP | 0.256 | 0.725 | | | | |
| EP | 0.712 | 0.173 | 0.769 | | | |
| EF | 0.207 | 0.493 | 0.348 | 0.729 | | |
| IQ | 0.570 | 0.209 | 0.703 | 0.357 | 0.870 | |
| PC | 0.243 | 0.619 | 0.134 | 0.376 | 0.150 | 0.730 |

| | CP | EP | EF | IQ | PC |
|---|---|---|---|---|---|
| | | | $Model_{Common}$ | | |
| CP | 0.778 | | | | |
| EP | 0.468 | 0.731 | | | |
| EF | 0.649 | 0.516 | 0.733 | | |
| IQ | 0.192 | 0.480 | 0.275 | 0.825 | |
| PC | 0.569 | 0.420 | 0.632 | 0.148 | 0.734 |

### 4.2.2. Assessment of Inner Models

The main results of the assessment of the inner models included hypothesis tests, determination coefficients, predictive relevance, and model fit for the three models.

### Hypothesis Tests

Based on the bootstrapping process, the program tested the hypotheses among the groups for the three models depending on the standardized errors. The proposed model was statistically significant, with most of the proposed relationships having a *p*-value less than 0.05, as shown in Figure 6 and Table 6, where insignificant relationships are indicated by red arrows.

**Table 6.** Path coefficients and *p*-value of the hypotheses for the three models.

| Model | Symbol | Hypothesis | Path Coefficient | *p*-Value | Conclusion |
|---|---|---|---|---|---|
| $Model_{client-consultant}$ | H1 | $CP \rightarrow CD$ | 0.31 | 0 | Supported |
| | H2 | $CP \rightarrow EP$ | 0.208 | 0.13 | Unsupported |
| | H3 | $EP \rightarrow CD$ | 0.219 | 0 | Supported |
| | H4 | $EF \rightarrow CP$ | 0.536 | 0 | Supported |
| | H5 | $EF \rightarrow CD$ | 0.171 | 0 | Supported |
| | H6 | $EF \rightarrow EP$ | 0.172 | 0.293 | Unsupported |
| | H7 | $EF \rightarrow IQ$ | 0.296 | 0.012 | Supported |
| | H8 | $EF \rightarrow PC$ | 0.571 | 0 | Supported |
| | H9 | $EF \rightarrow SQ$ | 0.551 | 0 | Supported |
| | H10 | $IQ \rightarrow CD$ | 0.144 | 0 | Supported |
| | H11 | $IQ \rightarrow EP$ | 0.298 | 0 | Supported |
| | H12 | $PC \rightarrow CD$ | 0.394 | 0 | Supported |
| | H13 | $PC \rightarrow EP$ | 0.034 | 0.774 | Unsupported |
| | H14 | $SQ \rightarrow CP$ | 0.307 | 0 | Supported |
| | H15 | $SQ \rightarrow CD$ | 0.076 | 0 | Supported |
| | H16 | $SQ \rightarrow EP$ | 0.152 | 0.117 | Unsupported |
| | H17 | $SQ \rightarrow IQ$ | −0.001 | 0.995 | Unsupported |
| | H18 | $SQ \rightarrow PC$ | 0.157 | 0.174 | Unsupported |

**Table 6.** *Cont.*

| Model | Symbol | Hypothesis | Path Coefficient | *p*-Value | Conclusion |
|---|---|---|---|---|---|
| *Model$_{contractor}$* | H1 | $CO \rightarrow CP$ | 0.161 | 0.441 | Unsupported |
| | H2 | $CO \rightarrow CD$ | 0.270 | 0 | Supported |
| | H3 | $CO \rightarrow EP$ | 0.484 | 0 | Supported |
| | H4 | $CO \rightarrow IQ$ | 0.534 | 0.103 | Unsupported |
| | H5 | $CO \rightarrow PC$ | 0.083 | 0.491 | Unsupported |
| | H6 | $CP \rightarrow EP$ | −0.114 | 0.374 | Unsupported |
| | H7 | $CP \rightarrow IQ$ | −0.023 | 0.902 | Unsupported |
| | H8 | $CP \rightarrow PC$ | 0.556 | 0 | Supported |
| | H9 | $EP \rightarrow CD$ | 0.316 | 0 | Supported |
| | H10 | $EF \rightarrow CO$ | 0.207 | 0.22 | Unsupported |
| | H11 | $EF \rightarrow CP$ | 0.460 | 0.002 | Supported |
| | H12 | $EF \rightarrow CD$ | 0.408 | 0.001 | Supported |
| | H13 | $EF \rightarrow EP$ | 0.164 | 0.295 | Unsupported |
| | H14 | $EF \rightarrow IQ$ | 0.286 | 0.102 | Unsupported |
| | H15 | $EF \rightarrow PC$ | 0.085 | 0.611 | Unsupported |
| | H16 | $IQ \rightarrow EP$ | 0.392 | 0.021 | Supported |
| | H17 | $PC \rightarrow CD$ | 0.426 | 0.000 | Supported |
| | H18 | $PC \rightarrow IQ$ | −0.073 | 0.697 | Unsupported |
| *Model$_{common}$* | H1 | $CP \rightarrow CD$ | 0.299 | 0.000 | Supported |
| | H2 | $CP \rightarrow EP$ | 0.189 | 0.051 | Unsupported |
| | H3 | $CP \rightarrow IQ$ | 0.037 | 0.709 | Unsupported |
| | H4 | $EP \rightarrow CD$ | 0.252 | 0.000 | Supported |
| | H5 | $EF \rightarrow CP$ | 0.482 | 0.000 | Supported |
| | H6 | $EF \rightarrow CD$ | 0.222 | 0.000 | Supported |
| | H7 | $EF \rightarrow EP$ | 0.215 | 0.053 | Unsupported |
| | H8 | $EF \rightarrow IQ$ | 0.285 | 0.032 | Supported |
| | H9 | $EF \rightarrow PC$ | 0.632 | 0.000 | Supported |
| | H10 | $IQ \rightarrow CD$ | 0.146 | 0.000 | Supported |
| | H11 | $IQ \rightarrow EP$ | 0.366 | 0.033 | Supported |
| | H12 | $PC \rightarrow CP$ | 0.264 | 0.002 | Supported |
| | H13 | $PC \rightarrow CD$ | 0.388 | 0.000 | Supported |

For the client–consultant model, the results revealed that the scope quality, contractual procedure, project characteristics, information quality, external factors, and estimator performance had a substantial effect on cost deviation. In addition, the scope quality and external factors had an indirect influence on cost deviation with contractual procedures, having *p*-values less than 0.001. Furthermore, external factors had an indirect influence on cost deviation with information quality, project characteristics, and scope quality, having path coefficients of 0.296, 0.57, and 0.551, respectively. Information quality had a considerable influence on estimator performance, which affects cost deviation.

On the other hand, the influence of the contractual procedure, project characteristics, external factors, and scope quality on estimator performance was insignificant, with a *p*-value above 0.05. The scope quality did not influence the information quality, estimator performance, or project characteristics, as shown in Table 6.

The procedure to evaluate whether the insignificant relationships of the client–consultant model should be retained is to study the outer loading and outer weights of factors with their group by comparing their value with the significance level chosen for this study (5%). The *p*-value of the outer weights and loading of all factors with their groups was less than 0.001. Hence, the relationships between the factors and their group are significant and the outer

model and inner model are linked. Consequently, the insignificant relationships among the groups were retained in the model.

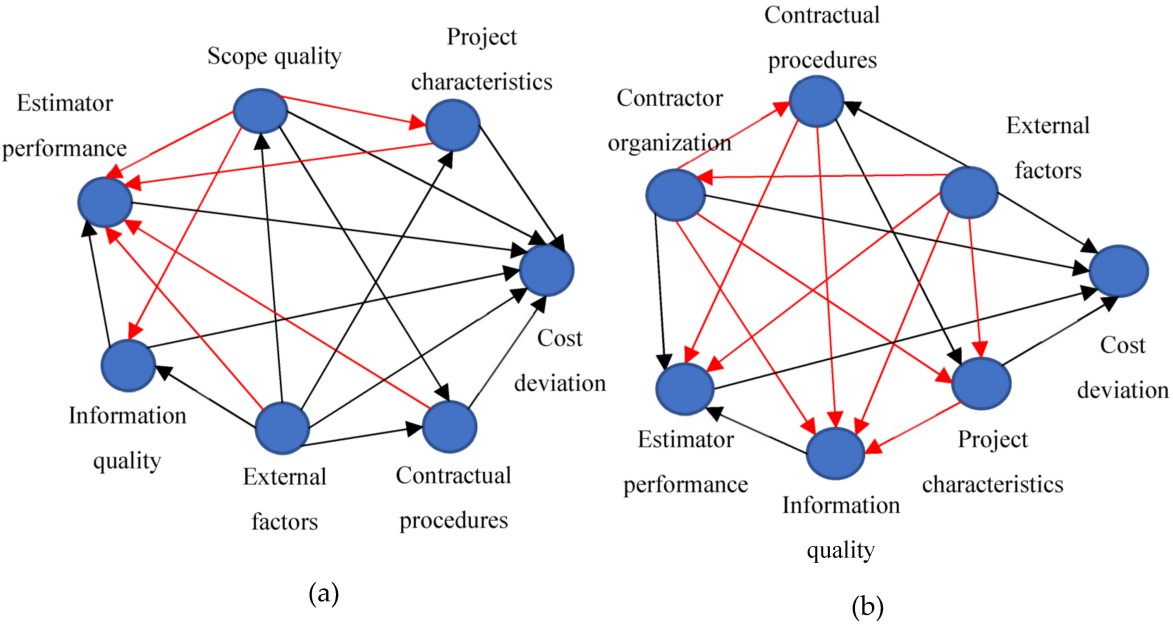

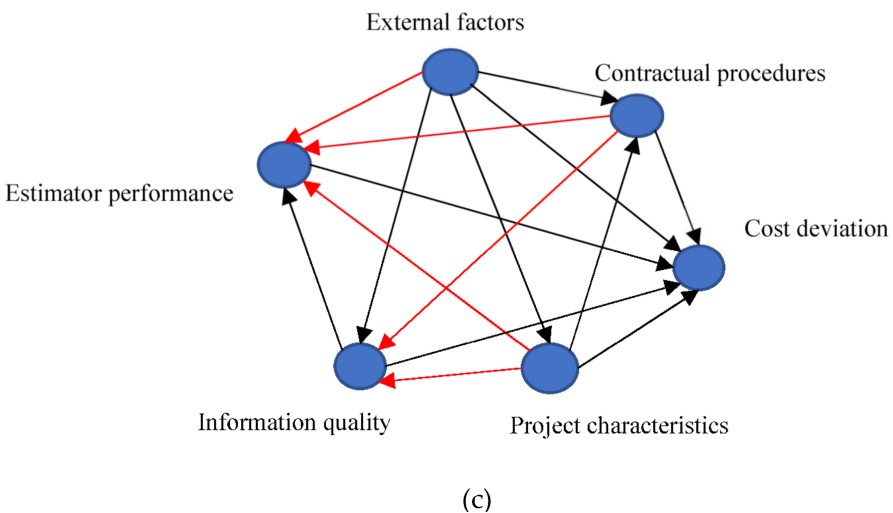

**Figure 6.** The three models: (**a**) client–consultant model, (**b**) contractor model, and (**c**) common model (insignificant relationships are indicated by red arrows).

For the common model, similar to the client–consultant model, the contractual procedures, estimator performance, external factors, information quality, and project characteristics had a significant influence on cost deviation, with path coefficients of 0.299, 0.252, 0.222, 0.143, and 0.388, respectively. In addition to the direct influence of external factors, these factors also had an indirect influence on cost deviation with contractual procedures, information quality, and project characteristics. The common model revealed a significant effect of information quality on the contractual procedure. Furthermore, the project characteristics affected cost deviation due to their influence on the contractual procedure.

Similar to the client–consultant model, the common model showed that contractual procedures, project characteristics, and external factors did not influence estimator performance, as shown in Table 6. In addition, information quality was not affected by project characteristics and contractual procedures. To study the linkage between the external

model and the internal model, the outer loadings and weight loadings of factors with their groups were assessed by the examination of their *p*-value with a significance level of 5%. The results revealed that the outer loading and weight loading were significant. Therefore, the outer and inner models were linked and the insignificant relationships, such as CP => EP, CP => IQ, EF => EP, PC => EP, and PC => IQ, were retained in the model.

The analysis of the contractor model revealed that the contractor's organization, contractual procedure, information quality, estimator performance, and project characteristics had an impact on cost deviation. Information quality and contractor organization had only an indirect influence on cost deviation by possessing an effect on estimator performance with a path coefficient of 0.484, where the relationship between contractor organization and information quality or project characteristics was insignificant. Furthermore, unlike the client–consultant and common models, the contractor model showed that there was an insignificant effect of external factors on contractor organization, estimator performance, information quality, and project characteristics. Therefore, external factors had only an indirect influence on cost deviation by affecting the contractual procedure. In addition, the contractual procedure did not affect the estimator performance or information quality. By studying the value of the outer loadings and weight loadings, there were some insignificant values.

Determination Coefficients

According to the assessment of the outer model, information quality had high outer loading values. It satisfied convergent validity without the elimination of any of its factors. These results were accepted, as the majority of participants reported a high dependency on historical data during construction cost estimation (18.5% of participants relied on historical data with a percentage of 90–100%, and 50.87% of participants relied on historical data with a percentage of 70–89%).

The findings of the three models showed that the contractual procedure and estimator performance played mediating roles between external factors and cost deviation as well as a relationship between information quality and cost deviation. In addition, the client–consultant and contractor models supported the mediation role of information quality and project characteristics in the relationships of the external factors with cost deviation: the $R^2$ and $R^2_{adj}$ of information quality were 0.087 and 0.07, respectively, and 0.45 and 0.44, respectively, for project characteristics, as shown in Table 7.

**Table 7.** $R^2$ and $R^2_{adj}$ of the groups for the three models.

| Group Symbol | $Model_{Client-consultant}$ | | $Model_{Contractor}$ | | $Model_{Common}$ | |
|---|---|---|---|---|---|---|
| | $R^2$ | $R^2_{adj}$ | $R^2$ | $R^2_{adj}$ | $R^2$ | $R^2_{adj}$ |
| SQ | 0.304 | 0.297 | | | | |
| CO | | | 0.043 | 0.027 | | |
| CP | 0.563 | 0.555 | 0.268 | 0.242 | 0.464 | 0.457 |
| PC | 0.450 | 0.440 | 0.396 | 0.364 | 0.400 | 0.396 |
| EP | 0.403 | 0.374 | 0.657 | 0.633 | 0.233 | 0.228 |
| EF | | | | | | |
| IQ | 0.087 | 0.070 | 0.391 | 0.347 | 0.076 | 0.070 |

The combined effect of external factors and scope quality on the contractual procedure was found to be significant in the client–consultant model, as shown in Figure 6a. Similarly, the influence of project characteristics and external factors was significant for the contractual procedure in the common model, as shown in Figure 6c. On the other hand, there was no double impact among the groups in the contractor model.

There were two insignificant sequences in the client–consultant model, including project characteristics with scope quality and scope quality with estimator performance, and two in the contractor model, including project characteristics with contractual procedures

and contractual procedures with information quality. However, there were no problems in the common model. This may be interpreted by the structure of the interaction between the groups. The client–consultant and contractor models may support the complex mediating effects among the different groups more than the direct relationships.

In addition to the results showing that all factor groups have a direct impact on CD, the findings of the three PLS-SEM models also showed that each model's indirect effects are different. For example, while the client–consultant model's indirect effects are SQ, EF, and IQ, the common model's indirect effects are (EF, IQ, and PC), and the contractor model's indirect effects are CO and IQ, respectively. These results show that the factors with the highest rankings are the information quality factors because they have both direct and indirect effects on cost accuracy from the viewpoints of both the client or consultant and the contractor.

For all the three models, external factors had a direct and indirect influence on cost deviation. However, it had no predictive relevance for cost deviation, in agreement with previous studies [8,29–31] reporting that external factors are time-consuming and have a direct influence on construction cost.

Predictive Relevance

After carrying out the hypothesis tests and calculating the determination coefficients of the three models, the importance of the groups for model forecasting was evaluated to determine the prediction relevance of each group. The findings related to $Q^2$ criteria, found in Table 8, were acceptable and reflected a good model with acceptable predictive relevance. The $Q^2$ coefficients for both external factors were not generated by the software because $Q^2$ values are provided only for endogenous latent variables. In general, the contractual procedure, project characteristics, estimator performance, and information quality had prediction relevance, and contractual procedure and information quality had the highest and lowest value in $Q^2$, respectively. The client–consultant model gave a higher $Q^2$ value for the four groups than in the common model, while the contactor model had the lowest value of the four groups. Therefore, clients and consultants give the four groups more importance and priority when assessing their impact on cost deviation than contractors. To compare $Q^2$ values of the common model with the previous studies, the $Q^2$ of the model developed by [8] for IQ, CP, and PC were 0.6, 0.371, and 0.249, respectively. There were differences in the value of the common model in this study; it may be attributed to the different location of the study and construction system between Saudi Arabia and Bahrain.

**Table 8.** Predictive relevance of the groups for the three models.

|  | SQ | CO | CP | PC | EP | EF | IQ |
|---|---|---|---|---|---|---|---|
| $Model_{Clint\ consultant}$ | 0.198 |  | 0.335 | 0.229 | 0.197 |  | 0.037 |
| $Model_{Contractor}$ |  | 0.020 | 0.118 | 0.177 | 0.292 |  | 0.155 |
| $Model_{Common}$ |  |  | 0.272 | 0.205 | 0.212 |  | 0.039 |

Model Fit

The goodness-of-fit (*GoF*) method was used to assess the PLS model [32]; it is the geometric mean of the average AVE and the average $R^2$, as shown in Equation (4).

$$GoF = \sqrt{\overline{AVE} \times \overline{R^2}} \tag{4}$$

The *GoF* depends on the performance of the outer and inner models. Akter et al. [33] categorized *GoF* values as large (*GoF* > 0.36), medium (0.1 < *GoF* < 0.36), and small (*GoF* < 0.1). Based on Tables 4 and 7, the *GoF* for the client–consultant, contractor, and common models was 0.46, 0.44, and 0.41, respectively.

## 5. Conclusions

Cost estimation errors can have detrimental impacts on construction projects, including project delays or cancellations, scope reductions, and significant financial risks for both owners and contractors. This study aimed to identify significant factors and investigate the impact of the interaction between these factors on cost deviation in the pre-tender stage of building projects in the Kingdom of Saudi Arabia. After collecting 57 factors affecting cost deviation from the literature, a questionnaire survey was conducted for contractors, consultants, and clients. Then, three PLS-SEM models (client–consultant, contractor, and common) were constructed. The study's findings showed that the issue affects both owners and contractors in terms of the project characteristics (function of the building, type of structures, quality of finishing, and duration of the project period), contractual procedures (tendering duration, amount of specialist work, and performance bond and warranty arrangements), and estimator performance (allowed for preparing cost estimates, number of estimating staff, and estimator's workload during estimation). The results also demonstrated that the issue of cost deviation is more significant to owners than contractors, with the client–consultant model showing a predictive relevance for project characteristics, contractual procedures, and estimator performance of 0.117, 0.118, and 0.292, respectively, for the contractor model. Beside the direct influence of all factor groups on CD, the finding revealed that the indirect influences are different for the three models, where SQ, EF, and IQ have indirect influence on CD for the client–consultant model, while the indirect influences for the common model and contractor model are (EF, IQ, and PC) and (CO, and IQ), respectively. To enhance the efficiency of cost deviation reduction, these results suggest that the factors with the highest rankings should be controlled, especially the factors of information quality, because they have direct and indirect influences on cost deviation from the client/consultant's views and the contractor's views.

**Funding:** The author extends his appreciation to the Deputyship for Research & Innovation, "Ministry of Education" in Saudi Arabia for funding this research work through the project number IFKSUHI-000.

**Institutional Review Board Statement:** Not applicable.

**Informed Consent Statement:** Not applicable.

**Data Availability Statement:** Common academic databases and search engines were utilized to find the research articles and journals that were used to create this work. The corresponding author will make raw data that support the conclusions of this study available upon request.

**Acknowledgments:** The author would like to thank all participants for answering the survey in this study.

**Conflicts of Interest:** The author declares no conflict of interest.

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
