# Peer review of "Cost Deviation Model of Construction Projects in Saudi Arabia Using PLS-SEM"

_sustainability, doi:10.3390/su142416391_

Round 1

Reviewer 1 Report

The authors need to following revisions:

1. To more clearly point out the research gaps of exsiting studies.

2. To give the in-depth explaination of the results by linking with real world construction projects.

3. To more clearly indicate the theoritical contributions of the study.

4. To more clearly present the practical contributions of the study.

Reviewer 2 Report

The author does a good job in describing the problem statement and the methodology used is well described as well. However, the reviewer has a few areas of concern that may strengthen the quality of the manuscript if addressed adequately. 1. There are a few minor spelling mistakes that need correction throughout the manuscript. (e.g. 'satistical' in figure 2. The author is highly encouraged to perform a spell check to rectify any errors in the manuscript. 2. While the motivation of the study is clear, the research gaps have not been adequately addressed. There have been previous studies that have already discussed the issue of cost overruns on projects. Just using a different methodology does not warrant the new study can be effectively used by the practitioners unless the author can provide compelling reasons or gaps that highlight the usage of the current PLS-SEM model. The author is encouraged in this regard to first highlight the existing gap in the literature and then strengthen the literature review section by adding further pertinent content. 3. Following directly the above point, the author is encouraged to modify the Results and Conclusions part based on the additional information gathered towards addressing the above point. 4. The 'Results and Discussion' section in the paper currently addresses just the 'Results' aspect. The author is encouraged to delineate the 'Discussion' section comparing the results to previous studies and how these findings can be of practical use to industry practitioners needs to be included. This will yield enhanced credibility to the practical usage of the model described in this manuscript. 5. The references will need to be uniformly included with regards to the font specifications. That is, uniformity in the font characteristics is desired. The reviewer commends the author for performing the current work and wishes the best to address the review comments provided herewith.

Reviewer 3 Report

In rows 106-111 there is similar text which is repeated twice,

Pease check EF numbers in table 3 and also in row 326

The last sentance of the Conclusions should probably be more informative (precise). Consider adding some text with more explanations of the results.

Round 2

Reviewer 1 Report

I am sastisfied with the revisions.

Reviewer 2 Report

The reviewer thanks the author for the good work in making the changes suggested in the previous round. I find the revised manuscript has been incorporated with useful content that lends better credibility to the research body. The reviewer feels the revised manuscript is ready to proceed to the next round of publication.